# Induction of G2/M Cell Cycle Arrest via p38/p21^Waf1/Cip1^-Dependent Signaling Pathway Activation by Bavachinin in Non-Small-Cell Lung Cancer Cells

**DOI:** 10.3390/molecules26175161

**Published:** 2021-08-25

**Authors:** Jih-Tung Pai, Ming-Wei Hsu, Yann-Lii Leu, Kuo-Ting Chang, Meng-Shih Weng

**Affiliations:** 1Division of Hematology and Oncology, Tao-Yuan General Hospital, Ministry of Health and Welfare, Taoyuan City 33004, Taiwan; jihtungpai@gmail.com; 2Department of Nutritional Science, Fu Jen Catholic University, New Taipei City 24205, Taiwan; namicola12@gmail.com; 3Graduate Institute of Natural Products, College of Medicine, Chang Gung University, Taoyuan City 33302, Taiwan; ylleu@mail.cgu.edu.tw; 4Tissue Bank, Chang Gung Memorial Hospital, Linkou, Taoyuan City 33342, Taiwan; 5Translational Medicine Center, Taoyuan General Hospital, Ministry of Health and Welfare, Taoyuan City 33004, Taiwan; kklkimonster@gmail.com

**Keywords:** non-small-cell lung cancer (NSCLC), bavachinin, G2/M cell cycle arrest, p21^Waf1/Cip1^, p38 MAPK

## Abstract

Lung cancer is the most commonly diagnosed malignant cancer in the world. Non-small-cell lung cancer (NSCLC) is the major category of lung cancer. Although effective therapies have been administered, for improving the NSCLC patient’s survival, the incident rate is still high. Therefore, searching for a good strategy for preventing NSCLC is urgent. Traditional Chinese medicine (TCM) are brilliant materials for cancer chemoprevention, because of their high biological safety and low cost. Bavachinin, which is an active flavanone of *Proralea corylifolia* L., possesses anti-inflammation, anti-angiogenesis, and anti-cancer activities. The present study’s aim was to evaluate the anti-cancer activity of bavachinin on NSCLC, and its regulating molecular mechanisms. The results exhibited that a dose-dependent decrease in the cell viability and colony formation capacity of three NSCLC cell lines, by bavachinin, were through G2/M cell cycle arrest induction. Meanwhile, the expression of the G2/M cell cycle regulators, such as cyclin B, p-cdc2^Y15^, p-cdc2^T161^, and p-wee1, was suppressed. With the dramatic up-regulation of the cyclin-dependent kinase inhibitor, p21^Waf1/Cip1^, the expression and association of p21^Waf1/Cip1^ with the cyclin B/cdc2 complex was observed. Silencing the p21^Waf1/Cip1^ expression significantly rescued bavachinin-induced G2/M cell accumulation. Furthermore, the expression of p21^Waf1/Cip1^ mRNA was up-regulated in bavachinin-treated NSCLC cells. In addition, MAPK and AKT signaling were activated in bavachinin-added NSCLC cells. Interestingly, bavachinin-induced p21^Waf1/Cip1^ expression was repressed after restraint p38 MAPK activation. The inhibition of p38 MAPK activation reversed bavachinin-induced p21^Waf1/Cip1^ mRNA expression and G2/M cell cycle arrest. Collectively, bavachinin-induced G2/M cell cycle arrest was through the p38 MAPK-mediated p21^Waf1/Cip1^-dependent signaling pathway in the NSCLC cells.

## 1. Introduction

Malignant tumor-caused death is a major health threat in the world, especially lung cancer. Histologically, lung cancer is categorized into small-cell lung cancer (SCLC) and non-small-cell lung cancer (NSCLC). Among them, about 80 to 85% of lung cancer patients are diagnosed with NSCLC [1]. Although, many clinical therapies for NSCLC patients have been administrated to increase the survival rate, the incident rate is still not improved, due to a lack of anticipation. Furthermore, more than 50% of NSCLC patients are identified as the advanced stage, when the cancer is initially diagnosed at their first diagnosis [2]. Therefore, searching for a good strategy for preventing tumorigenesis might be a way to lower the incident rate of lung cancer.

Dysregulation of the cell cycle progression is a crucial feature of the malignant cells. The cell cycle is divided into the G1, S, G2, and M (mitosis) phase. Each of the cell cycle phases is governed by the cyclins and cyclin-dependent kinases (CDKs). For example, the cyclin D/CDK4 and cyclin E/CDK6 complexes regulate the cell from G1 to entering the S phase; cyclin A/CDK2 and cyclin A/cdc2 (CDK1) control the G2 phase cells into the M phase. In addition, cyclin B/cdc2 is a major complex that guides M-phase progression [3]. Moreover, the activities of the cyclin/CDK complexes are regulated by the cyclin-dependent kinase inhibitors (CDKIs) and tumor suppressor protein, p53. CDKIs are divided into INK4 and CIP/KIP families. The members of the INK4 family are p15^INK4b^, p16^INK4a^, p18^INK4c^, and p19^INK4d^, which suppress the activities of CDK4 and CDK6 [4]. The p21^Waf1/Cip1^, p27^Kip1^, and p52^Kip2^ belong to the CIP/KIP family, which controls the cell cycle progression by broadly interacting with specific cyclin/CDK complexes [5]. These negative regulators of the cell cycle are generally categorized as tumor suppressors.

Alteration of the cell cycle regulator expressions in lung carcinogenesis has been demonstrated. Overexpression of cyclin D1 and loss of p16^INK4a^ expression are observed in early lung tumorigenesis, and accounted for 40–50% of NSCLC patients [6,7]. A high level of cyclin D1 and reduced expression of p16^INK4a^ in lung cancer patients show the worst prognosis [8]. Additionally, low-level expression of the KIP/CIP family of CDKIs accounts for about 30–65% of NSCLC [9]. Loss of p21^Waf1/Cip1^ expression is regularly detected in the advanced stage of lung cancer [10]. Analysis of the relationship between CDKIs expression and survival rate reveals that both the inactivation of p21^Waf1/Cip1^ and p16^INK4a^ in lung cancer patients display a brief overall survival [11]. Meanwhile, high expression of cyclin B1 and CDK1 has been observed in the early stage of lung cancer [12,13]. Thus, targeting cell cycle regulators might be a strategy for lung cancer therapies and prevention.

Traditional Chinese medicine (TCM) possesses high safety and low-cost properties, and is an extremely interesting material for anti-cancer drug development and cancer chemoprevention in clinical research. *Proralea corylifolia* L. is a well-known TCM, and has been used for skin disease, diuretic disease, and inflammatory disease treatment in Eastern Asia countries for many years. Until now, at least 20 active components have been reported from seed extracts. Most of them belong to flavonoid, coumarin, furanocoumarin, and chalcone [14]. Bavachinin, an active flavanone in *Proralea corylifolia* L. seed extract, has been reported to possess anti-inflammation, anti-angiogenesis, anti-cancer, and anti-hepatic steatosis activities [15,16,17,18,19]. Molecularly, bavachinin promotes hypoxia-inducible factor-1α (HIF-1α) degradation, and led to the suppression of vascular endothelial cell growth factor (VEGF)-regulated angiogenesis [15]. Furthermore, the induction of p53-regulated intrinsic apoptosis, by bavachinin, has also been demonstrated in dimethylhydrazine-induced colon tumor, in vivo [18]. Although, bavachinin-induced apoptosis in A549 lung cancer cells and G2/M cell cycle arrest in SCLC cells, through peroxisome proliferator-activated receptor γ (PPARγ)/reactive oxygen species (ROS) and the ATM/ATR signaling pathway, has been established, respectively [16,20]. The precisely anti-lung cancer mechanisms of bavachinin are still unidentified, especially in NSCLC. In the present study, the anti-cancer molecular mechanisms of bavachinin, in NSCLC cells, were exposed. The current results revealed that the induction of p21^Waf1/Cip1^-mediated G2/M cell cycle arrest, via the p38 MAPK signaling pathway, might be the target of bavachinin-treated NSCLC cells. Our results might provide an opportunity for NSCLC prevention.

## 2. Results

### 2.1. The Cytotoxic Effect of Bavachinin on Non-Small-Cell Lung Cancer Cells

To interpret the cytotoxicity activity of bavachinin (Figure 1A) on NSCLC cells, three NSCLC cell lines were inspected. The cells were treated with serial concentrations of bavachinin, for an indicated time interval. Afterwards, the cell viability was scrutinized, as indicated in the Material and Methods section. As shown in Figure 1B, the cell viabilities of the A549, H23, and HCC827 cells were decreased in dose- and time-dependent modes. However, the cell viabilities of A549 and HCC827 were slightly induced in 10 μM of bavachinin incubation for 24 h. The cytotoxicity effects of bavachinin on normal lung epithelial cells were also measured. The results revealed that the repression of 16HBE cell viability was only detected after 30 μM of bavachinin treatment for 48 h and 72 h, respectively (Figure 1B). Furthermore, inhibiting the colony formation of three NSCLC cell lines, by bavachinin, was obviously perceived in a dose-dependent manner (Figure 1C). These results showed that the cytotoxic activity of bavachinin was more potent in lung cancer cells than normal cells.

### 2.2. Induction of G2/M Cell Cycle Arrest in Bavachinin-Inhibited Non-Small Cell Lung Cancer Cell Viability

To investigate the possibility that the cell viability, suppressed by bavachinin, results from cell cycle disruption, flow cytometric analyses were performed. As shown in Figure 2, dose-dependent G2/M-phase cell accumulation was observed in the bavachinin-added NSCLC cell lines, except the H23 cells. The maximum accumulation of the G2/M phase in H23 cells was revealed in 20 μM, rather than 30 μM bavachinin treatment (Figure 2B). In the A549 cells, the accumulation of the G2/M cell population was increased from 17.64 ± 2.4% to 63.70 ± 0.52%, after 30 μM of bavachinin treatment for 24 h (Figure 2A). About 46%, 23%, and 30% of the G2/M cell population was accumulated in 30 μM of bavachinin-treated A549, H23, and HCC827 cells, respectively (Figure 2).

### 2.3. Induced G2/M Cells Accumulation via Bavachinin through p21^Waf1/Cip1^-Mediated Signaling Pathway

The molecular mechanisms of bavachinin-induced G2/M cell cycle arrest in NSCLC cells were established. Initially, the cyclin B/cdc2 complex, which plays an important role in G2/M cell cycle regulation, was studied. The expressions of cyclin B, p-cdc2^Y15^, and p-cdc2^T161^ were dose-dependently decreased after bavachinin incubation in A549 cells (Figure 3A). Furthermore, the expression of p-weel was also down-regulated, in a dose-dependent mode, in bavachinin-treated A549 cells. However, the up-regulation of p21^Waf1/Cip1^, which is the CDK inhibitor, was clearly discovered after bavachinin addition. Meanwhile, the markers expression in bavachinin-added H23 and HCC827 cells were examined. Consistently, the expression of the molecular profiles in the bavachinin-treated H23 and HCC827 cells was similar in the A549 cells with bavachinin treatment (Figure 3B).

To verify the role of p21^Waf1/Cip1^ in bavachinin-induced G2/M cell cycle arrest, the expression of p21^Waf1/Cip1^ was silenced by siRNA technology, and then cell cycle distribution was noticed. As shown in Figure 3C, bavachinin-induced p21^Waf1/Cip1^ expression was suppressed by p21^Waf1/Cip1^ siRNA transfection. Moreover, bavachinin-induced G2/M cell cycle arrest was dramatically blocked by p21^Waf1/Cip1^ siRNA transfection (Figure 3C). The results implied that p21^Waf1/Cip1^ might play a crucial role in bavachinin-induced G2/M cell cycle arrest in NSCLC cells. To explore the function of p21^Waf1/Cip1^ in bavachinin-mediated cell cycle arrest, immunoprecipitation analyses were used to examine the interaction of p21^Waf1/Cip1^ with the cyclin B/cdc2 complex. The results exposed that the association of p21^Waf1/Cip1^ with cyclin B and/or p-cdc2^T161^ was increased by bavachinin treatment in three types of NSCLC cell lines (Figure 3D). To further assess the characters of bavachinin-induced p21^Waf1/Cip1^ expression, via transcriptional the mechanism-dependent pathway, p21^Waf1/Cip1^ mRNA expression was explored. A dose-dependent up-regulation of p21^Waf1/Cip1^ mRNA expression was perceived in bavachinin-treated A549 cells (Figure 3E). Similarly, up-regulation of p21^Waf1/Cip1^ mRNA expression was also noticed in 30 μM of bavachinin-treated H23 and HCC827 cells (Figure 3F).

### 2.4. Up-Regulation of p21^Waf1/Cip1^ Expression by Bavachinin through p38 MAPK-Mediated Signaling Pathway

Bavachinin-induced G2/M cell cycle accumulation, through p21^Waf1/Cip1^ transcriptional-dependent mechanisms, was established (Figure 3). Recent studies indicated that the regulation of p21^Waf1/Cip1^ expression, throughout the MAPKs and AKT-dependent signaling pathway, is shown [21,22,23,24]. To understand the molecular mechanisms of bavachinin-regulated p21^Waf1/Cip1^ expression, the MAPKs and AKT signaling pathways were studied. A549 cells were incubated with serial concentrations of bavachinin for 24 h. Afterwards, the activation of MAPKs and AKT were probed by Western blot. The expressions of p-ERK, p-JNK, p-p38 MAPK, and p-AKT were dose-dependently increased in bavachinin-added A549 cells (Figure 4A). Furthermore, the effects of bavachinin on the MAPKs and AKT activation in H23 and HCC827 cells were inspected. Up-regulations of phosphorylated proteins of MAPKs and AKT were also perceived after 30 μM of bavachinin treatment in H23 and HCC827 cells, except the expression of p-AKT, which was suppressed in bavachinin-added HCC827 cells (Figure 4B). Specific pharmacological inhibitors were applied to ascertain the functions of the signaling molecules in bavachinin-induced p21^Waf1/Cip1^ expression. The outcomes showed that the repression of p38 MAPK activation, by a pharmacological inhibitor, dramatically restrained bavachinin-induced p21^Waf1/Cip1^ expression in three NSCLC cell lines (Figure 4C).

The pharmacological inhibitor of p38 MAPK, SB203580, was applied, to address the role of p38 MAPK in bavachinin-induced G2/M cell cycle arrest regulation. The results revealed that bavachinin-induced p38 MAPK activation was repressed by SB203580 pretreatment. The up-regulation of p21^Waf1/Cip1^ expression, by bavachinin, was also inhibited in SB203580-pretreated A549 cells (Figure 5A). In addition, bavachinin-induced p21^Waf1/Cip1^ mRNA expression was suppressed by SB203580 (Figure 5B). Restrained bavachinin-accumulated G2/M population cells were observed in SB203580 and bavachinin co-treatment (Figure 5C). Meanwhile, rescuing bavachinin-inhibited cell viability was detected in the co-treatment group (Figure 5D).

## 3. Discussion

A high mutation rate in the cell cycle regulators has been reported in NSCLCs [9]. Therefore, targeting the cell cycle regulators may be a strategy for lung cancer prevention and treatment. The restraint of lung cancer proliferation via bavachinin, which is the active component of *Proralea corylifolia* L., was assessed in the present study. The results revealed that the cell viability of three NSCLC cell lines were repressed by bavachinin treatment. However, the inhibitory effect of cell viability on the human bronchial epithelial cell line, 16HBE, was only perceived in 30 μM of bavachinin treatment, for more than 48 h. The analyses of cell cycle distribution showed that a significant increase in the G2/M cell population was discovered in three NSCLC cell lines. Moreover, bavachinin induced the expression of p21^Waf1/Cip1^, and increased the p21^Waf1/Cip1^ and cyclin B/cdc2 complex association. Silencing the expression of p21^Waf1/Cip1^ repressed bavachinin-induced G2/M cell accumulation. In addition, bavachinin-induced p21^Waf1/Cip1^ expression, via p38 MAPK-regulated transcriptional mechanisms, was recognized.

The mammalian cell cycle is intensely monitored by four stages of G1, S, G2, and M phases. The G1 and G2 phases represent gap periods for cells preparing to complete the S and M phase, respectively. As the cells prepare to transverse into the S phase, the complexes of D-type cyclins plus CDK4/6, and the complex of cyclin E/CDK2, are activated, and then enable cells to enter the S phase for DNA syntheses. After DNA replication, the cells move to the G2 phase and prepare for the M phase. The expression of the cyclin A/cdc2 (CDK1) complex is decreased at the G2 phase, and follows cyclin B/cdc2 complex activation for promoting chromosome condensation and progression through the M phase [25]. The phosphorylation of cdc2 is a crucial step to regulate the cyclin B/cdc2 complex activity. The phosphorylation status of cdc2 at Tyr15 and Thr161 are important sites to regulate cyclin B/cdc2 complex activation. After dephosphorylated cdc2 interacted with cyclin B, p-weel is immediately phosphorylated at Tyr15 on cdc2. Meanwhile, CDK-activating kinase (CAK) is phosphorylated at Thr161. In the interim, the activity of the cyclin B/cdc2 complex is inhibited. Thereafter, the phosphorylated Tyr15 on cdc2 is hydrolyzed by Cdc25 phosphatase, when the cells progress to mitosis. Meantime, the phosphorylated Thr161 is maintained, to trigger cyclin B/cdc2 complex activation. The dephosphorylation of Thr161 is initiated, and then cyclin B and cdc2 are degraded [26], after mitosis completion. Thus, the suppression of cyclin B/cdc2 complex expressions and/or disruption of the cdc2 phosphorylation status, could arrest cells at the G2/M phase. In the present study, the anti-proliferation ability of bavachinin was inspected in three NSCLC cell lines, A549, HCC827, and H23 cells. The results exposed that the cell viabilities of the three NSCLC cells were blocked in dose- and time-dependent modes. In addition, the ability of colony formation was clearly decreased by bavachinin treatment, in a dose-dependent manner (Figure 1). Furthermore, dramatic accumulation of G2/M cells was perceived in bavachinin-treated NSCLC cells (Figure 2). Analyzing the cyclin B/cdc2 signaling pathway indicated that the expressions of cyclin B, p-cdc2^Y15^, and p-cdc2^T161^ were decreased after bavachinin incubation. Moreover, the expression of p-weel was also inhibited by bavachinin, in three NSCLC cells (Figure 3A,B). Thus, the results implied that bavachinin-induced G2/M cell cycle arrest might occur through inhibiting p-weel expression and promoting cyclin B and cdc2 protein degradation, followed by restrained cyclin B/cdc2 activity. Interestingly, bavachinin-induced G2/M cell cycle arrest in NSCLC cells was similar to bavachinin-suppressed SCLC cell proliferation [20]. These outcomes inferred that bavachinin might be qualified for lung cancer prevention, no matter whether it is NSCLC or SCLC.

CDKIs are another important factor to govern the cell cycle processes. The p21^Waf1/Cip1^, which is a member of the INK4 family, is a critical CDKI, to regulate the G1 and G2/M cell cycle. The up-regulation of p21^Waf1/Cip1^ accumulates cancer cells at the G1 or G2/M phase. The enhancement of the interaction of p21^Waf1/Cip1^ with the cyclin B/cdc2 complex has demonstrated the cell accretion at the G2/M phase [27]. The outcomes showed that the treatment with bavachinin dramatically induced the protein level of p21^Waf1/Cip1^ in three NSCLC cell lines (Figure 3A). Moreover, the relationship between bavachinin up-regulated p21^Waf1/Cip1^ and G2/M cell cycle arrest was explored. The results showed that the bavachinin-accumulated G2/M cell population was rescued before p21^Waf1/Cip1^ knockdown by siRNA transfection (Figure 3B). Moreover, immunoprecipitation analyses indicated that the association of p21^Waf1/Cip1^ with cyclin B and/or p-cdc2^T161^ was increased in bavachinin-incubated lung cancer cells (Figure 4A). The consequences strongly implied that bavachinin-induced G2/M cell cycle arrest was, at least in part, through the p21^Waf1/Cip1^-dependent signaling pathway.

Protein activities may be measured by transcriptional regulation and/or post-translational modification. It is known that the induction of p21^Waf1/Cip1^ expression is governed by p53-mediated transcriptional regulation in the cell cycle [5]. Nevertheless, the p53-independent up-regulation of p21^Waf1/Cip1^ gene expression has been verified [21,28]. The mechanism of bavachinin-induced p21^Waf1/Cip1^ expression, via transcriptional regulation, was examined. As shown in Figure 3B, a dose-dependent increase in p21^Waf1/Cip1^ mRNA expression was observed in bavachinin-treated A549 cells. Furthermore, up-regulation of p21^Waf1/Cip1^ mRNA expression was also detected in bavachinin-treated H23 and HCC827 cells. The results indicated that bavachinin-induced p21^Waf1/Cip1^ expression occurred through transcriptional mechanisms. Interestingly, the up-regulation of p21^Waf1/Cip1^ expression was detected in both p53 wild-type (A549) and p53 mutant-type (H23 and HCC827) cells, after bavachinin treatment [29,30]. We speculated that bavachinin-induced p21^Waf1/Cip1^ expression might occur through the p53-independent signaling pathway. However, the possibility of a p53-regulated p21^Waf1/Cip1^ signaling pathway was not excluded in bavachinin-incubated A549 cells.

Numerous studies indicate that the MAPK and PI3K/AKT signaling pathways are important mechanisms to regulate cancer proliferation [22,31,32]. Stress-induced cell cycle arrest has also been displayed through the MAPK-regulated p21^Waf1/Cip1^ signaling pathway [21,23]. Therefore, the expression of the kinases was examined in bavachinin-induced cell cycle arrest. The results exposed that the expressions of phosphorylated JNK, p38 MAPK, and AKT were increased in bavachinin-treated cells (Figure 4A). Additionally, the administration of the specific pharmacological inhibitors, to clear the role of the kinases in the regulation of p21^Waf1/Cip1^ expression, was executed. The outcomes disclosed that only the pre-treatment of specific p38 MAPK inhibitors reversed bavachinin-induced p21^Waf1/Cip1^ expression in the three NSCLC cell lines (Figure 4C). The evaluation of p38 MAPK and bavachinin-induced p21^Waf1/Cip1^-mediated G2/M cell cycle arrest was carried out. The results disclosed that bavachinin-induced p38 MAPK activation and p21^Waf1/Cip1^ expression were suppressed by p38 MAPK inhibitor pre-treatment (Figure 5A). Besides, the repression of bavachinin-induced p21^Waf1/Cip1^ mRNA expression was also perceived in p38 MAPK inhibitor-pretreated A549 cells (Figure 5B). The outcomes of cell cycle distribution revealed that the induction of G2/M arrest, by bavachinin, was rescued before p38 MAPK inhibitor pre-treatment. In the meantime, the increase in G1 population cells was noted in the p38 MAPK inhibitor and bavachinin co-treatment cells (Figure 5C). This upsurge of G1 population cells was caused by the p38- MAPK inhibitor rescuing the bavachinin-induced G2/M cells.

*Psoralea corylifolia* L. has been used as folk medicine in Ayurveda and Chinese medicines for many years. The analyses of the biological components of *Psoralea corylifolia* exhibited that the major constitutes are flavones, flavonoids, coumarins, chalcones, and monoterpenes [33]. These active components have been designated to anti-Alzheimer, anti-oxidant, anti-diabetic, and anti-cancer activities [14]. Many active compounds have been recognized against breast, prostate, liver, and lung cancer activities [34,35,36,37,38,39]. Bavachinin, a prenylflavone in the seed of *Psoralea corylifolia*, has been indicated as a natural PPARγ agonist. Structurally, the isopentenyl group and the methoxy group of the A ring have critical effects on the PPARγ agonist activity of bavachinin (Figure 1A) [40]. Therefore, bavachinin has been directed against lung cancer cells proliferation, by targeting the PPARγ ROS-regulated signaling pathway [16]. Triggering PPARγ activation, by p38 MAPK, has been reported in transforming growth factor β (TGFβ)-induced epithelial mesenchymal transition (EMT) in NSCLC cells [41]. These studies imply that the p38 MAPK-regulated PPARγ signaling pathway may play a pivotal role in NSCLC cell tumorigenesis. However, bexarotene-induced apoptosis, via the activation of the PPARγ-mediated signaling pathway, is also demonstrated in NSCLC cells [42]. Accordingly, the role of the PPARγ signaling pathway in NSCLC tumorigenesis is still controversial. The present results showed that bavachinin-induced G2/M cell cycle arrest might occur through the p38 MAPK-mediated p21^Waf1/Cip1^ signaling pathway in NSCLC cells. The role of PPARγ in our system should be scrutinized in the future. Although our study showed the critical role of p38 MAPK in p21^Waf1/Cip1^-regulated G2/M cell cycle arrest, in bavachinin-treated NSCLC cells, the activation mechanism of p38 MAPK-regulated p21^Waf1/Cip1^ is still mysterious and worthy to be explored. Furthermore, the present study only focused on investigating G2/M cell cycle arrest regulation. Cell viability suppression, resulting from bavachinin-induced apoptotic mechanisms, could not be excluded in the present system. Further evaluation of these anti-cancer abilities, and underlying molecular mechanisms of bavachinin, should be conducted. According to the results, bavachinin-induced G2/M cell cycle arrest might be, in part, through the p38 MAPK-mediated p21^Waf1/Cip1^ signaling pathway in lung cancer cells. Bavachinin might provide a potent candidate for a therapeutic and/or chemo-preventive agent for NSCLC.

## 4. Materials and Methods

### 4.1. Chemicals and Reagents

Bavachinin was purchased from MedChemExpress (South Brunswick Township, NJ, USA). Anti-p-cdc^Y151^, p-cdc^T161^, p-21^Waf1/Cip1^, p-weel^S642^, cyclin B1, p-ERK, ERK, JNK, anti-p-p38 and anti-p38 antibodies were obtained from Cell Signaling (Beverly, MA, USA). Anti-p-JNK, anti-β-actin and protein A/G plus agarose were acquired from Santa Cruz Biotechnology (Santa Cruz, CA, USA). Further, 3-(4,5-dimethylthiazol-2-yl)-2,5-diphenyl tetrazolium bromide (MTT) was took from Merck KGaA (Darmstadt, Germany).

### 4.2. Cell Culture and Cell Viability Assay

The NSCLCs, A549, H23, and HCC827 cell lines were purchased from the American Type Culture Collection (Manassas, VA, USA). The human bronchial epithelial cell line, 16HBE14o- (16HBE), was kindly provided by Dr. Kuo-Ting Chang (Taoyuan General Hospital). All the NSCLCs and 16HBE cells were incubated in 5% fetal bovine serum-containing RPMI-1640 (Hyclone Laboratories, Logan, UT, USA) and cultured in a 5% CO_2_ atmosphere at 37 °C. Cells were seeded on concentration as 1 × 10^4^/well in 96-well plates for 24 h. Subsequently, indicated concentrations of bavachinin (0, 10, 20, and 30 μM) were supplemented for 24, 48, and 72 h, respectively. The cell viability was evaluated by MTT method.

### 4.3. Colony Formation Assay

Three NSCLC cell lines were seeded (1 × 10^3^/well) on 6-well plates in triplicate for 24 h. Designated concentrations of bavachinin (0, 10, 20, and 30 μM) were added for 24 h. After bavachinin incubation, complete culture media were replaced every two days until 13 days. After that, culture media were discarded. The culture was washed and fixed with 70% ice methanol for 30 min. The 0.1% of crystal violet was added for colony staining after cell fixation. Colonies were washed, removing residue crystal violet, photographed, and then quantitated by Quantity One system (Bio-Rad Laboratories, Hercules, CA, USA).

### 4.4. Cell Cycle Distribution Analysis

The NSCLC cells were seeded (5 × 10^5^/well) in 6 cm Petri dish and treated by indicated concentrations of bavachinin (0, 10, 20, and 30 μM) for 24 h. Cells were collected, washed, and fixed by 70% ice ethanol thereafter. The fixed cells were then centrifuged and washed by 1 × PBS twice. After the washing, resuspended cells were incubated with 1 μL of Triton-X100, 2 μL of RNase (0.5 mg/mL) and 10 μL of propidium iodide (1 mg/mL, Sigma Chemical, St. Louis, MO, USA) for 30 min of cell staining. The DNA contents were detected by FACScan laser flow cytometer (Beckman Coulter, Fullerton, CA, USA).

### 4.5. Western Blot Analysis

Bavachinin-treated cell lysates were extracted by RIPA buffer. After cell lysates quantification, sodium dodecyl sulfate-polyacrylamide gel electrophoresis (SDS-PAGE) was performed to separate the protein samples. Thereafter, SDS-PAGE gels were transferred to the PVDF membrane (Millipore, Burlington, MA, USA). The transferred membranes were blocked by 3% skim milk for 30 min, and then hybridized with indicated primary antibodies at 4 °C overnight. After hybridization, membranes were washed by washed buffer and soaked with secondary antibody at room temperature for 1 h. The protein expressions were analyzed by chemiluminescence (ECL kit, Amersham Pharmacia Biotech, Chicago, IL, USA). The intensities of protein were quantitated by a UVP BioSpectrum Imaging System ChemiDoc-It2 810 (UVP, LLC, Upland, CA, USA). The level of β-actin was used as the internal control.

### 4.6. Immunoprecipitation

The immunoprecipitation method was analyzed by modified protocols from Liao et al. [43]. Briefly, bavachinin-treated cell lysates (1 mg of total protein concentration) were precleared with protein-A/G agarose bead at 4 °C for 20 min. After that, the supernatant was collected and incubated with anti-p21^Waf1/Cip1^ antibody at 4 °C for 2 h. The mixtures were then reacted with rabbit anti-mouse IgG for the secondary precleared process at 4 °C for 30 min. The immune complexes were incubated with protein-A/G agarose bead overnight, and then washed by immunoprecipitation buffer and resuspended with protein loading buffer-containing RIPA extraction buffer. The samples were next for Western blot analyses for indicated protein expression detection.

### 4.7. Quantitative Reverse-Transcription Polymerase Chain Reaction

After bavachinin treatment, total RNA was extracted from the cells using TRI Reagent^®^ (Merck KGaA, Darmstadt, Germany). Reverse transcription was then performed using the GoScript RT RT-PCR kit (Promega, Madison, WI, USA). Real-time qPCR was used to investigate the expression of p21^Waf1/Cip1^, glyceraldehyde 3-phosphate dehydrogenase (GAPDH) as the internal control gene. Each reaction was performed according to the protocols of SYBR Green qPCR Master Mix kit (Thermo Fisher Scientific, Taipei, Taiwan). The RT-qPCR reaction conditions were set as 95 °C for 15 sec, and 60 °C for 60 sec for 40 cycles after 95 °C for 5 min pre-modification. The forward primer of p21^Waf1/Cip1^ was as follows: 5′-ATGTCCGTCAGAACCCATGC-3′ and the reverse primer of p21^Waf1/Cip1^ was as follows: 5′-TCGAAGTTCCATCGCTCACG-3′, while the GAPDH forward primer sequences were as follows: 5′-CTCTGGTAAAGTGGATATTGT-3′ and reverse primer was as follows: 5′-GGTGGAATCATATTGGAACA-3′. The relative expression of each gene was calculated using the 2^−ΔΔCq^ method. The average of three experiments each performed in triplicate with SEs was presented.

### 4.8. Small Interfering RNA Transfection

A549 cells were cultured to 70% confluence and then transfected with the p21^Waf1/Cip1^ siRNA duplexes by using GenMuteTM siRNA Transfection Reagent (SignaGen Laboratories, Ijamsville, MD, USA) according to the manufacturer’s instructions. Briefly, siRNA duplexes were mixed and reacted with GenMuteTM siRNA Transfection Reagent for 15 min. After the reaction, the siRNA mixtures were gently added to A549 cells. After transfection, cellular levels of the p21^Waf1/Cip1^ proteins specific for the siRNA transfection were checked by immunoblotting. All experiments were performed after transfection at 6 h.

### 4.9. Statistical Analysis

The results were expressed as the mean ± SD calculated from at least three independent determinations. One-way ANOVA coupled with Dunnett’s *t* tests were used to compare individual experiments with the control. A probability of *p* < 0.05 was considered to be a significant difference. 

## Figures and Tables

**Figure 1 molecules-26-05161-f001:**
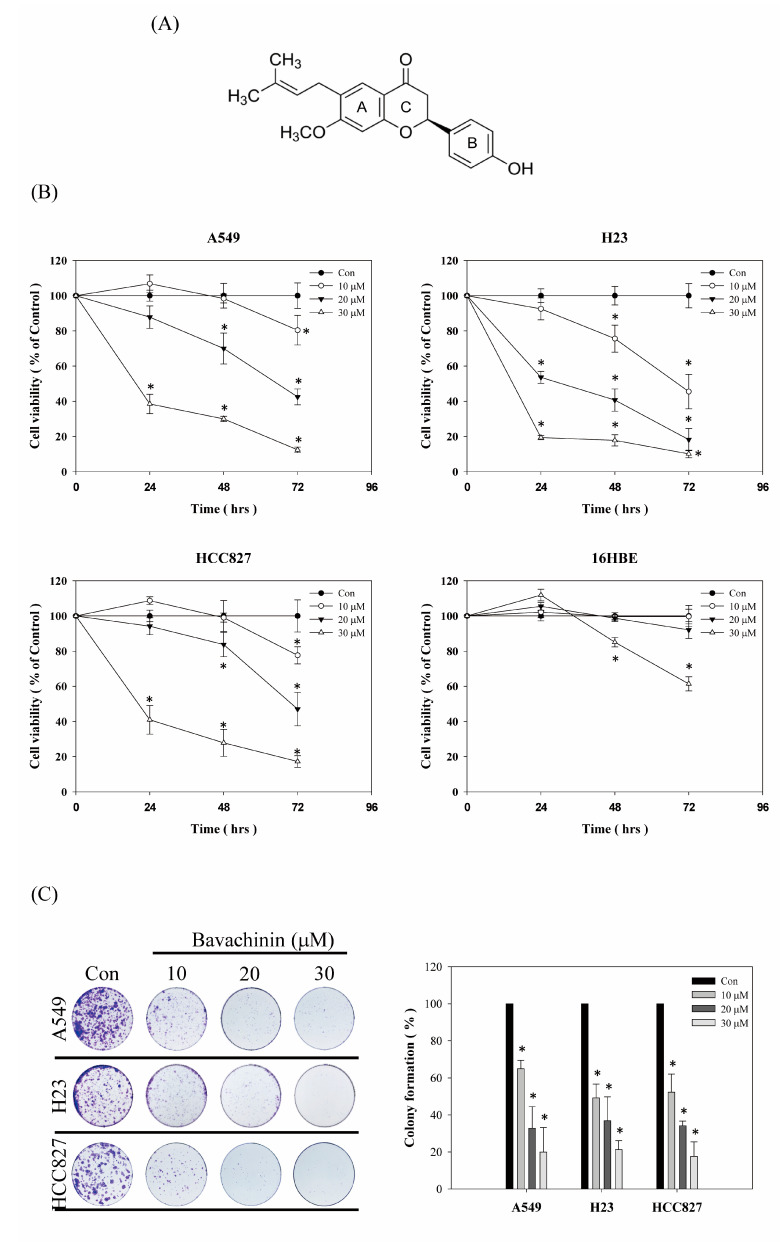
The cytotoxicity effects of bavachinin in NSCLC cells. (**A**) The chemical structure of bavachinin. (**B**) Three different NSCLC cells and 16HBE lung epithelial cells were seeded and incubated with bavachinin (0, 10, 20, 30 μM) for indicated time intervals. After incubation, cell viability was analyzed by MTT assay. (**C**) The NSCLC cells were seeded and stimulated with indicated concentrations of bavachinin (0, 10, 20, and 30 μM) for 24 h. Thereafter, bavachinin-containing media were replaced to complete culture medium every two days for 13 days. The culture was then washed, fixed, and stained with crystal violet. Colonies were counted as described in the Material and Methods. All results were at least triplicated and represented by the mean ± S.D. The statistically significant difference was labeled as * when *p* < 0.05 between control and experimental group.

**Figure 2 molecules-26-05161-f002:**
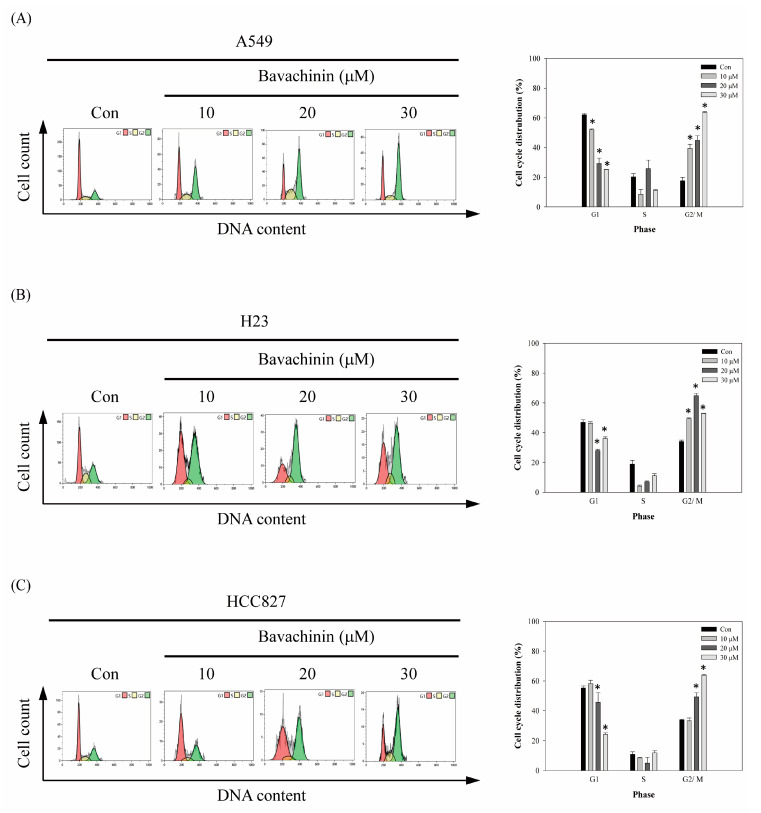
The effects of bavachinin on cell cycle distribution of NSCLC cells. The (**A**) A549, (**B**) H23, (**C**) HCC827 cells were plated and stimulated with the serial dosages of bavachinin for 24 h. Afterwards, cells were harvested and stained to analyze the cell cycle distribution by flow cytometry. All results were triplicated and represented by the mean ± S.D. The statistically significant difference was labeled as * when *p* < 0.05 between the control and the experimental group.

**Figure 3 molecules-26-05161-f003:**
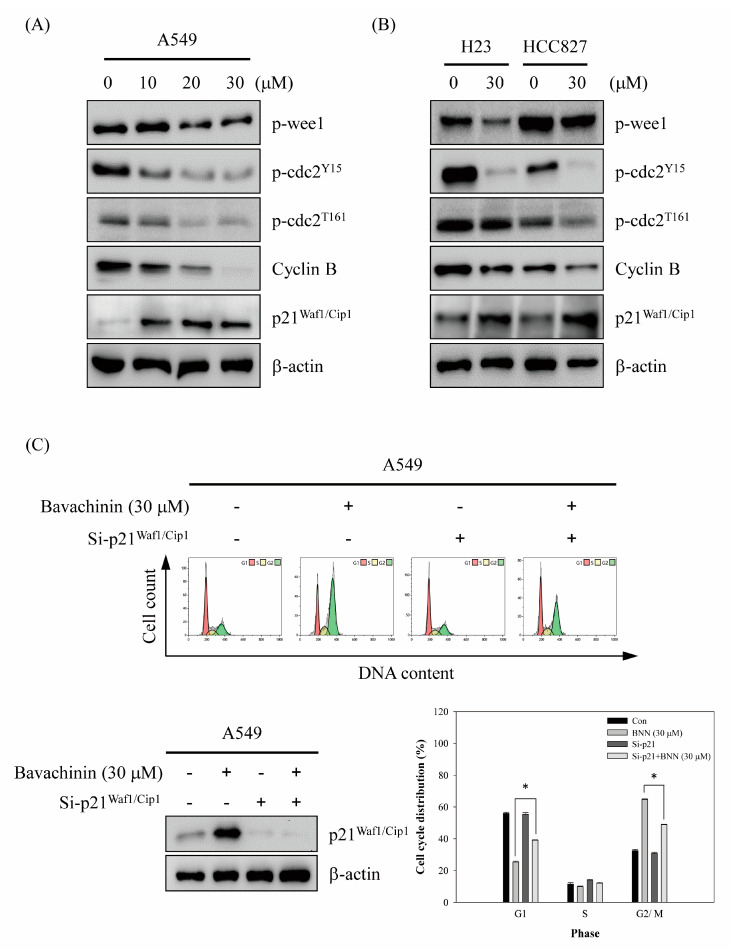
The effects of bavachinin on G2/M cell cycle regulator expressions of NSCLC cells. (**A**) A549, (**B**) H23 and HCC827 cells were cultured and incubated with indicated concentrations of bavachinin for 24 h. Subsequently, the expressions of p-wee1, p-cdc2^Y15^, p-cdc2^T161^, cyclin B and p21^Waf1/Cip1^ were analyzed by Western blot. (**C**) A549 cells were transfected by p21^Waf1/Cip1^ siRNA, and then incubated with 30 μM of bavachinin for 24 h. Next, the expression of p21^Waf1/Cip1^ and cell cycle distribution was examined by Western blot and flow cytometry, respectively. (**D**) Three different NSCLC cells were treated with 30 μM of bavachinin for 24 h, and then immunoprecipitation analyses were performed as described in the Materials and Methods. The interaction of p-cdc2^T161^, cyclin B, and p21^Waf1/Cip1^ were then assayed by Western blotting. (**E**,**F**) Three different NSCLC cells were treated with indicated concentrations of bavachinin for 24 h, the expressions of p21^Waf1/Cip1^ mRNA were then analyzed by quantitative RT-PCR. The results were represented by the mean ± S.D. The statistically significant difference was labeled as * when *p* < 0.05 between the control and the experimental group.

**Figure 4 molecules-26-05161-f004:**
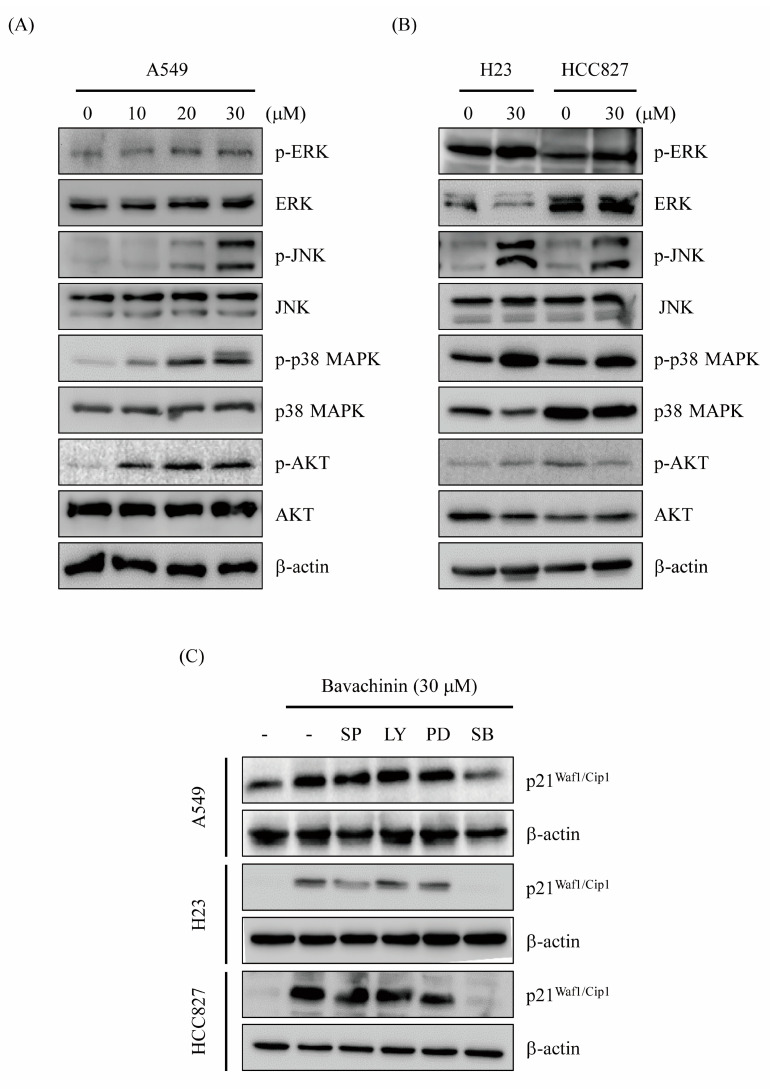
The effects of bavachinin on MAPK and AKT signaling pathway. (**A**) A549, (**B**) H23 and HCC827 cells were cultured and incubated with indicated concentrations of bavachinin for 24 h. Thereafter, the protein levels of phosphorylated and dephosphorylated forms of ERK, JNK, p38 MAPK and AKT were investigated. (**C**) A549, H23 and HCC827 cells were pre-treated with specific pharmacological inhibitors (JNK inhibitor: SP600125 (SP); AKT inhibitor: LY294002 (LY); ERK inhibitor: PD98059 (PD); p38 MAPK inhibitor: SB203580 (SB)) for 1 h before stimulated with 30 μM of bavachinin for 24 h. The expression of p21^Waf1/Cip1^ was then inspected by Western blot.

**Figure 5 molecules-26-05161-f005:**
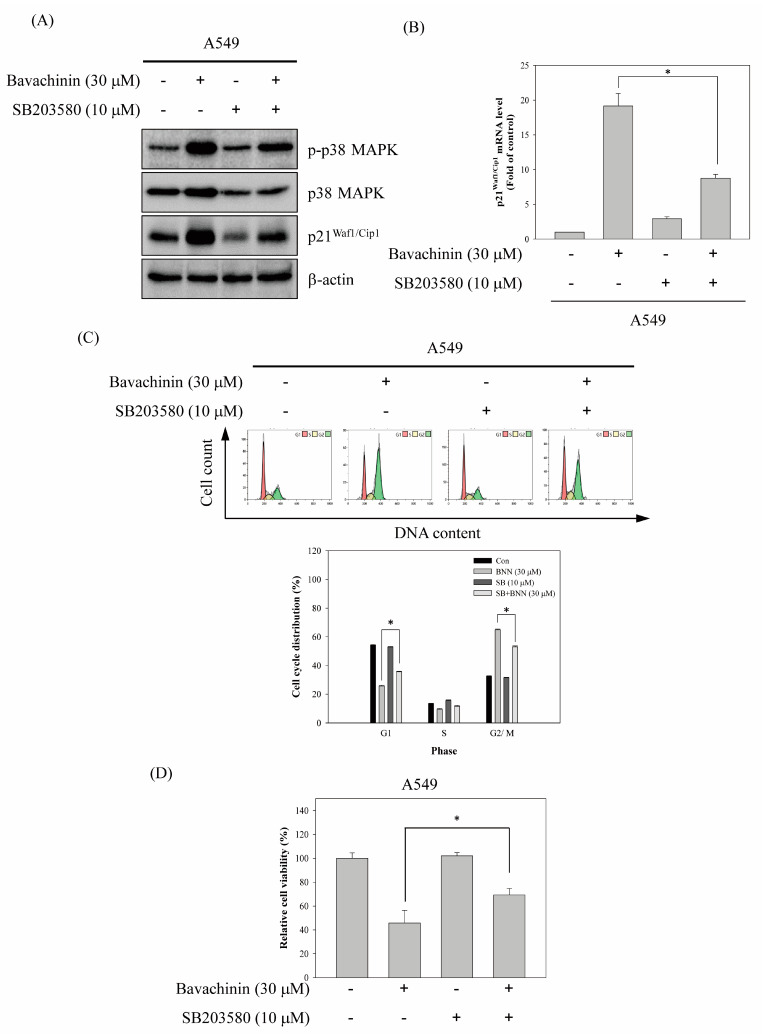
p38 MAPK-p21^Waf1/Cip1^ signaling pathway involved in bavachinin-induced G2/M cell cycle arrest. A549 cells were pre-treated with or without 10 μM of SB203508 for 1 h before bavachinin incubation for 24 h. After treatment, (**A**) the protein expressions of p-p38 MAPK, p38 MAPK, p21^Waf1/Cip1^, and β-actin, and (**B**) the mRNA expressions of p21^Waf1/Cip1^, (**C**) the cell cycle distributions, and (**D**) the cell viability were determined. All results were triplicated and represented by the mean ± S.D. The statistically significant difference was labeled as * when *p* < 0.05 between the control and the experimental groups.

## Data Availability

The data presented in this study are available in this article.

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
