# Peer review of "Induction of G2/M Cell Cycle Arrest via p38/p21Waf1/Cip1-Dependent Signaling Pathway Activation by Bavachinin in Non-Small-Cell Lung Cancer Cells"

_molecules, 2021, doi:10.3390/molecules26175161_

Round 1

Reviewer 1 Report

Summary

The authors evaluated the activity of bavachinin on cell cycle and cell viability in non-small cell lung cancer. They found that bavachinin inhibited cell survival through induction of cell cycle arrest at the G2/M phase. They further investigate the molecular mechanism of bavachinin mediated cell cycle arrest and identify the p38/p21 signaling pathway to be key downstream effector for bavachinin’s anti-cancer properties in non-small cell lung cancer.

Comments

This manuscript is logically written, however there is need for a moderate level of editing to the English language. A few other general comments: 1. All figure legends and axis-labels need to be enlarged; 2. Stats were not always used through the study or clearly marked when used. All stats need to be indicated with a “*”; 3. Imbed the cell line used within each figure, not only in the figure legend; 4. Material and methods section needs additional details.

A549 and H23 are KRAS mut and HCC827 is EGFR mut. Do these oncogene addicted phenotypes have anything to do with the effect of bavachinin? Does the same cell cycle arrest occur in NSCLC cells that lack these driver mutations?

To further identify the molecular mechanism of bavachinin, the authors should also investigate downstream of p21 induced growth arrest. For example, these three cell lines are known to have high MYC or induce MYC upon drug resistance. Does bavachinin have any effects on MYC or other transcription factors regulated by the cell cycle or that are important for cell survival?  

Figure 1A is not adequately addressed. Either describe the importance of the structure for the study or remove it from the manuscript.

Figure 1B: There are no error bars on the control treatments, even if normalized to 100 there should still be error bars from being performed in triplicate.

Figure 1C: Add the time for colony formation to the figure or figure legend and add stats to quantification panel on the right.

Figure 2 identifies a G2/M cell cycle arrest with bavachinin treatment. The authors should also evaluate apoptosis as a mechanism of bavachinin inhibited cell viability. This could be through cell cycle arrest or an independent mechanism.

Figure 3: Add western blot of cell cycle regulators in normal lung epithelial cells since 30 uM does inhibit cell viability at 72 hrs. Indicated significance with * on bars in C. For IP experiments, there is not an IgG control. This needs to be added to ensure the antibody is selectively pulling down p21.

Figure 4 investigated the effects of bavaachinin on the MAPK and AkT signaling pathways, however this is very briefly described lines 173-183. The wording and reasoning are unclear here. There needs to be more explanation of experimental rationale and design. i.e., What does each drug inhibit and what are the treatment conditions?

Figure 4C: Label the targeted kinase within the figure for each drug and prove that each inhibitor is blocking the intended target like in 5a with SB and p38.

Figure 5 validates p38 as a druggable target upstream of p21 driven cell cycle arrest. However, the next logical step is to investigate the effect of p38 inhibition on cell viability. The authors should repeat figure 1 with bavachinin, SB or the combo.

Author Response

Reviewer 1 comments:

Question 1

This manuscript is logically written, however there is need for a moderate level of editing to the English language. A few other general comments: 1. All figure legends and axis-labels need to be enlarged; 2. Stats were not always used through the study or clearly marked when used. All stats need to be indicated with a “*”; 3. Imbed the cell line used within each figure, not only in the figure legend; 4. Material and methods section needs additional details.

Answer:

Thank you for the reviewer’s comments, we have enlarged all figure axis labels, added the symbols in the all results with statistically significant differences, embedded the names of cell lines within the figures, and detailed descriptions in the Materials and Methods section. Furthermore, we also re-edited the manuscript with English skills.

Question 2

A549 and H23 are KRAS mut and HCC827 is EGFR mut. Do these oncogene addicted phenotypes have anything to do with the effect of bavachinin? Does the same cell cycle arrest occur in NSCLC cells that lack these driver mutations?

Answer:

We appreciated the reviewer’s comments. Based on our information, the cell cycle disruption activity of bavachinin in more than one NSCLC line was firstly evaluated in the present study. As shown in Figure 2, bavachinin-induced G2/M cell cycle arrest was observed in three NSCLC lines, no matter KRAS or EGFR mutation (Figure 2). Accordingly, we speculated bavachinin-induced G2/M cell cycle arrest might not be correlated with these two oncogenes’ mutations. However, the anti-NSCLC molecular mechanisms of bavachinin are still mysteries and worthy to be explored. Additionally, further experiments are required to evaluate the relationship between bavachinin-induced G2/M cell cycle arrest and KRAS or EGFR mutation. We will assess the subject in our next study.

Question 3

To further identify the molecular mechanism of bavachinin, the authors should also investigate downstream of p21 induced growth arrest. For example, these three cell lines are known to have high MYC or induce MYC upon drug resistance. Does bavachinin have any effects on MYC or other transcription factors regulated by the cell cycle or that are important for cell survival? 

Answer:

Thank you very much for the reviewer’s constructive comments. The present study was evaluated the anti-NSCLC activity and underlying molecular mechanisms of bavachinin. For the first time, bavachinin-induced G2/M cell cycle arrest via p38-MAPK/p21Waf1/Cip1-dependent signaling pathway in NSCLC cells was indicated. However, there are limited literatures regarding the anti-cancer molecular mechanisms of bavachinin, especially in NSCLC cells. Many unknowns are still crucial and additional examinations are required. As we discussed in the manuscript, the regualtion of p21Waf1/Cip1 and other target molecules of bavachinin in anti-NSCLC mechanisms should be further verified. We sincerely appreciate that reviewer’s comment provide us a possible molecular target for further exploration of the anti-NSCLC mechanism by bavachinin. We will keep verifying the targets of bavachinin in our future study.  

Question 4

Figure 1A is not adequately addressed. Either describe the importance of the structure for the study or remove it from the manuscript. 

Answer:

We acknowledge the reviewer’s comment. We have added some descriptions of bavachinin structure in the Discussion, as described “Bavachinin, a prenylflavone in the seed of Psoralea corylifolia, has been indicated as a natural PPARg agonist. Structurally, the isopentenyl group and the methoxy group of A ring have the critical effects on the PPARg agonist activity of bavachinin (Figure 1A).” in page 14 , line 26 to page 15, line 3.

Question 5

Figure 1B: There are no error bars on the control treatments, even if normalized to 100 there should still be error bars from being performed in triplicate. 

Answer:

We thank the reviewer’s comment. We have added the error bar on the control groups in Figure 1B.

Question 6

Figure 1C: Add the time for colony formation to the figure or figure legend and add stats to quantification panel on the right.

Answer:

We have added the information in figure legend of Figure 1C as described “The NSCLC cells were seeded and stimulated with indicated concentrations of bavachinin (0, 10, 20, and 30 mM) for 24 h. Thereafter, bavachinin-containing media were replaced to complete culture medium every two days for 13 days. The culture was then washed, fixed, and stained with crystal violet. Colonies were counted as described in Material and Methods”. In addition, we also added a symbol of statistically significant difference in the right panel of Figure 1C.

Question 7

Figure 2 identifies a G2/M cell cycle arrest with bavachinin treatment. The authors should also evaluate apoptosis as a mechanism of bavachinin inhibited cell viability. This could be through cell cycle arrest or an independent mechanism.

Answer:

The anti-NSCLC activity and underlying molecular mechanism of bavachinin was evaluated. As shown in Figure 2, G2/M cell cycle arrest was observed in three NSCLC cell lines after bavachinin treatment. Furthermore, bavachinin-induced G2/M cell cycle arrest through p38-MAPK/p21Waf1/Cip1-dependent signaling pathway in NSCLC cells was indicated. The present study focused on the inspection of the molecular mechanisms of G2/M cell cycle arrest regulated by bavachinin. Thus, we didn’t appraise the induction of apoptotic ability of bavachinin in the current system. However, bavachinin suppressed cell viability via apoptotic induction could not be excluded in this study. We will validate the induction of apoptotic ability by bavachinin in NSCLC cells in our coming study. In addition, we also added the description “Furthermore, the present study only focused on investigating the G2/M cell cycle arrest regulation. Cell viability suppression resulting from bavachinin-induced apoptosis mechanisms could not be excluded in the present system. Further evaluation of these anti-cancer abilities and underlying molecular mechanisms of bavachinin should be conducted.” in page 15, line 17-21.

Question 8

Figure 3: Add western blot of cell cycle regulators in normal lung epithelial cells since 30 uM does inhibit cell viability at 72 hrs. Indicated significance with * on bars in C. For IP experiments, there is not an IgG control. This needs to be added to ensure the antibody is selectively pulling down p21.

Answer:

Inhibition of cell viability in three NSCLC cells was observed after 24 h of bavachinin treatment. However, the cell viability of 16HBE normal epithelial cells decreased about 15% and 39% after 30 mM of bavachinin treatment for 48 and 72 h, respectively. The cell viability of 16HBE cells was not affected after bavachinin incubation for 24 h (Figure 1B). The evaluation of the suppressing NSCLC cell viability by bavachinin resulted from cell cycle distribution was performed. The flow cytometry and Western blot technologies were applied after bavachinin addition for 24 h. It might be not necessary to inspect the expressions of cell cycle regulators in our system since cell viability of 16HBE cells was not affected after bavachinin treatment for 24 h. Meanwhile, we also added the symbol of significant difference in Figure 3C.

Non-specific antibody of the same isotype of p21Waf1/Cip1 capture antibody – IgG antibody to pre-clearing the sample was employed to avoid non-specific binding in IP experiments. The immunoprecipitation procedure was illustrated in Materials and Methods section in details (page 17, line 23 to page 18, line 6).

Question 9

Figure 4 investigated the effects of bavachinin on the MAPK and AkT signaling pathways, however this is very briefly described lines 173-183. The wording and reasoning are unclear here. There needs to be more explanation of experimental rationale and design. i.e., What does each drug inhibit and what are the treatment conditions?

Answer:

The descriptions of the experimental rationale and design about the effects of bavachinin on the MAPK and AKT signaling pathway were enhanced in page 9, line 8-22.

Question 10

Figure 4C: Label the targeted kinase within the figure for each drug and prove that each inhibitor is blocking the intended target like in 5a with SB and p38.

Answer:

The specific pharmacological inhibitors of MAPK and AKT were utilized to appraise the relationship between kinases and 21Waf1/Cip1 expression. The target kinases were identified in bavachinin-induced 21Waf1/Cip1 expression by the treatment of the well-known inhibitors for tumor biological research. As shown in Figure 4C, suppression of bavachinin-induced 21Waf1/Cip1 expression was only observed in p38-MAPK inhibitor and bavachinin co-treatment. The role of p38-MAPK in bavachinin-induced 21Waf1/Cip1 expression was displayed in Figure 5.

The bavachinin-induced 21Waf1/Cip1 expression was not attenuated by other inhibitors. Therefore, the link of these kinase and bavachinin-induced 21Waf1/Cip1 expression was disconnected. However, efficiencies of inhibitors on the target kinase expression by bavachinin treatment were determined in A549 cells. The results were listed below. ( The results  were list in the attached file) 

Furthermore, the detailed descriptions of the target kinase for each inhibitor were included in figure legends.

Question 11

Figure 5 validates p38 as a druggable target upstream of p21 driven cell cycle arrest. However, the next logical step is to investigate the effect of p38 inhibition on cell viability. The authors should repeat figure 1 with bavachinin, SB or the combo.

Answer:

The results of the cell viability were added in Figure 5D to address the question. The sentence was changed to “Meanwhile, rescuing bavachinin-inhibited cell viability was detected in the co-treatment group (Figure 5D).” in page 10, line 7-8.

Reviewer 2 Report

In this manuscript, the authors did a research about Induction of G2/M cell cycle arrest via p38/p21Waf1/Cip1-dependent signaling pathway activation by bavachinin in non-small cell lung cancer cells. The research idea is clear, and the writing meets the specifications. If the author can modify or explain the following issues, it is recommended to accept the manuscript. A Minor Revision is suggested.

  1. The author should provide bavachinin’s structural characterization information
  2. In cell colony formatting assay authors used a concentration of high cell inhibition activity, the decrease of colonies formatted may be due to cell death caused by compounds. The authors should prove that the concentration they used in this assay does not lead to a significant cell death.
  3. Increase of P21 may result in arrestment of G1 or G2 / M, but the results show only G2 / M arrestment, the author should explain the reason.
  4. Quantitative analysis of some important results, such as Figure 3c, should be done for statistical difference analysis.
  5. Exploring the P38 and P21 relationships using only small molecular inhibitors are not enough, and the experiment siRNA should be supplemented.

Author Response

Question 1

The author should provide bavachinin’s structural characterization information

Answer:

We thank the reviewer’s comment. We have added some descriptions of bavachinin structure in the Discussion, as described “Bavachinin, a prenylflavone in the seed of Psoralea corylifolia, has been indicated as a natural PPARg agonist. Structurally, the isopentenyl group and the methoxy group of A ring have the critical effects on the PPARg agonist activity of bavachinin (Figure 1A).” in page 14, line 26 to page 15, line 3.

Question 2

In cell colony formatting assay authors used a concentration of high cell inhibition activity, the decrease of colonies formatted may be due to cell death caused by compounds. The authors should prove that the concentration they used in this assay does not lead to a significant cell death.

Answer:

In the colony formation assay, bavachinin was treated for 24 h and then replaced with a complete culture medium every two days for 13 days. Afterward, media was discarded and the culture was washed, fixed, and stained with crystal violet for counting the colonies. And the results revealed that colony formation ability was dose-dependently suppressed by bavachinin in three NSCLC cells. Inhibition of cell viability by bavachinin was considerably displayed at 30 mM for 24 h treatment in A549 and HCC827 cells and 20 mM for 24 h in H23 cells. Thus, suppression of colony formation ability by a high concentration of bavachinin resulting from cell viability inhibition was not excluded. However, cytotoxicity in 10 mM of bavachinin tereatment for 24 h was not observed in three NSCLC cells (Figure 1B). Interestingly, treatment with 10 mM of bavachinin for 24 h and then replaced the complete culture medium, the colony formation ability was still obviously suppressed in three NSCLC cells (Figure 1C). Accordingly, repression of colony formation ability under low concentration (10 mM) of bavachinin treatment in three NSCLC cells was not resulted by cell viability inhibition.

Question 3

Increase of P21 may result in arrestment of G1 or G2 / M, but the results show only G2 / M arrestment, the author should explain the reason.

Answer:

As shown in Figure 3C, G2/M cell population was induced after bavachinin treatment. In addition, the G1 population cells were apparently decreased. No cell cycle disruption was monitored in silencing p21Waf1/Cip1 expression only. However, bavachinin-induced G2/M cell cycle arrest was suppressed by si-p21Waf1/Cip1 transfection. In the meantime, the G1 population cells were concomitantly increased (Figure 3C). Therefore, we suggested that this upsurge of G1 population cells was caused by p21Waf1/Cip1 down-regulation rescuing bavachinin-induced G2/M cells, rather than G1 arrest induction.

Question 4

Quantitative analysis of some important results, such as Figure 3c, should be done for statistical difference analysis.

Answer:

The analysis of statistically significant difference was accomplished in all Figures according to the reviewer’s suggestion.

Question 5

Exploring the P38 and P21 relationships using only small molecular inhibitors are not enough, and the experiment siRNA should be supplemented.

Answer:

It has been well-known that SB203580 is a specific inhibitor of p38-MAPK, the applicable approach to justify the relationship between p38-MAPK and p21Waf1/Cip1. Furthermore, bavachinin-induced p21Waf1/Cip1 mRNA and protein expression was suppressed by p38-MAPK inhibitor treatment in Figure 5. Meanwhile, bavachinin-induced G2/M cell cycle arrest and cytotoxicity were also rescued by p38-MAPK inhibitor treatment. The application of a small molecular inhibitor was enough to verify the relationship between the role of p38-MAPK in bavachinin-induced p21Waf1/Cip1 expression and cell cycle arrest. 

Round 2

Reviewer 1 Report

Overall, I am satisfied that the revised manuscript adequately addressed most concerns, and I recommend the manuscript for publication.